# Exploring Potential Intermediates in the Cross-Species Transmission of Influenza A Virus to Humans

**DOI:** 10.3390/v16071129

**Published:** 2024-07-14

**Authors:** Chung-Young Lee

**Affiliations:** 1Department of Microbiology, School of Medicine, Kyungpook National University, Daegu 41944, Republic of Korea; cylee87@knu.ac.kr; 2Untreatable Infectious Disease Institute, Kyungpook National University, Daegu 41944, Republic of Korea

**Keywords:** influenza A virus, cross-species transmission, zoonosis, intermediate host, pandemic

## Abstract

The influenza A virus (IAV) has been a major cause of several pandemics, underscoring the importance of elucidating its transmission dynamics. This review investigates potential intermediate hosts in the cross-species transmission of IAV to humans, focusing on the factors that facilitate zoonotic events. We evaluate the roles of various animal hosts, including pigs, galliformes, companion animals, minks, marine mammals, and other animals, in the spread of IAV to humans.

## 1. Introduction

Influenza A virus (IAV) is a significant pathogen that primarily causes respiratory illnesses in humans [1]. Since its identification, IAV has been recognized as a major pathogen responsible for several notable pandemics in the 20th and 21st centuries, including the 1918 Spanish flu, the 1957 Asian flu, the 1968 Hong Kong flu, and the 2009 H1N1 pandemic [1,2]. These events underscore the virus’s capacity to cause widespread illness and significant mortality, emphasizing its critical impact on global public health.

Wild waterfowl are considered natural reservoirs of IAV, as the most diverse subtypes of the virus have been detected in these birds [3]. The segmented genome of IAV allows for frequent genetic exchange between different virus strains, leading to substantial viral diversity. This genetic reassortment, particularly during the co-infection of different IAV subtypes in the same host, accelerates the emergence of novel IAVs capable of infecting new hosts [4]. For example, the 2009 H1N1 pandemic virus (pdm09) emerged through reassortment events between human, swine, and avian IAVs [2,4]. Pigs played a crucial role in the genesis of pdm09 and its transmission to humans, underscoring the significant role of intermediate hosts in the emergence of new pandemics.

Potential intermediate hosts, such as pigs, poultry, and other animals, may serve as bridge hosts in the cross-species transmission of IAVs [4,5,6,7,8]. Similar to the case of SARS-CoV-2, where unknown intermediate hosts were vital for the virus’s adaptation and spread in humans [9], IAVs also rely on intermediate hosts to bridge transmission from their natural reservoirs to humans [6,7,8]. This review explores potential intermediate hosts of IAVs and evaluates their roles in cross-species transmission, focusing on the factors that facilitate zoonotic events.

## 2. Essential Factors for Zoonotic Events of IAVs

### 2.1. The Nature of the Relationship between Animals and Humans

Since their initial domestication approximately 15,000 years ago, animals have maintained a multifaceted relationship with humans. This relationship extends beyond mere companionship, encompassing critical roles such as providing sustenance and serving as transportation partners. However, this close association also carries inherent risks, particularly when it comes to the transmission of pathogens. As agricultural practices have advanced, the crowding of animals in confined spaces—often driven by profit maximization—has fostered an environment conducive to rapid pathogen spread and evolution. This intensified interaction between animals and humans raises the potential for zoonotic spillover events [10,11,12].

Individuals who work closely with infected animals face heightened exposure to animal-origin pathogens [13]. While most zoonotic spillovers are dead-end events, occasional establishment of stable host–virus interactions occurs. Additionally, companion animals—those with both physical and emotional proximity to humans—may serve as conduits for pathogen transmission to their owners. In light of these complexities, assessing the degree of interaction between animals and humans becomes crucial for predicting and preventing zoonotic spillover events.

### 2.2. Receptor Binding Specificity

Receptor binding specificity is crucial for IAV attachment to novel host cells, determining its host range and transmissibility. The α2,3 and α2,6 sialic acid linkages are key determinants of IAV host specificity [14,15]. Traditionally, human-adapted IAVs preferentially bind to α2,6-linked sialic acids (α2,6-SA) found predominantly in the human upper respiratory tract, facilitating efficient transmission [16,17,18]. In contrast, avian IAVs bind to α2,3-linked sialic acids (α2,3-SA), which are prevalent in avian respiratory tracts. The structural differences in the hemagglutinin (HA) protein of IAV account for these specific binding preferences, which are major determinants of the virus’s ability to infect different host species. However, more recent studies revealed that humans also have α2,3-SA in their lower respiratory tracts and that avian species have varied distributions of α2,3- and α2,6-SA glycans [16,19,20,21]. Moreover, the diversity of sialic acid-containing glycans in the respiratory tract has complicated the understanding of receptor binding specificity for IAVs of different origins [22].

Specific mutations in the receptor binding domain (RBD) of the HA protein, such as E190D/G225D in H1 or Q226L/G228S in H2 and H3 HAs, can increase binding affinity for α2,6-SA, aiding the adaptation of avian IAVs to human hosts [7,23,24,25,26,27]. This enhances the virus’s ability to attach to and enter human respiratory epithelial cells, increasing its transmissibility. Additionally, an optimal balance between HA receptor-binding avidity and neuraminidase (NA) receptor-destroying activity is essential for efficient viral entry and budding, related to efficient transmission between humans [7,28,29]. NA cleaves sialic acids from glycoproteins, facilitating the release of newly formed virions from infected cells.

NA also functions as a secondary receptor binding protein, aiding in the initial attachment of the virus to host cells [30,31,32]. Traditionally known for its role in cleaving sialic acid residues to facilitate viral release, NA’s binding capabilities enhance viral attachment, particularly when HA binding is suboptimal. A second sialic acid-binding site (2SBS) is at the loops of 368–370 and 399–403 in the NA, and it interacts with sialic acid receptors [30,33]. The sequence of 2SBS is preserved in avian IAVs but missing in human or swine IAVs [30,32,34]. This dual functionality of NA contributes to the efficiency of viral entry and subsequent infection, emphasizing the importance of the HA-NA balance in viral transmission and adaptation.

### 2.3. Acid Stability and Viral Transmission

The acid stability of the HA protein is another critical factor influencing IAV transmission. Respiratory viruses, including IAV, are primarily transmitted through contact, fomites, or airborne routes [35]. Viral transmission via aerosolized droplets produced by speaking, coughing, or sneezing is particularly concerning, as these droplets can widely disperse and infect susceptible individuals at a distance [36].

Studies have shown that IAVs with increased acid stability are likely to be more airborne transmissible in humans [37,38,39,40,41]. In 2012, highly pathogenic avian H5N1 IAVs that are capable of droplet transmission in ferrets were experimentally generated during adaptation in the ferret animal model [42,43]. Notably, these mutant viruses acquired mutations located in the vicinity of the fusion peptide in HA, increasing the acid and thermal stability of the mutant viruses, which is thought to enable droplet transmission of the mutant viruses in ferrets [42,43,44]. The environment of the human respiratory tract, which is slightly acidic (pH 5.5–6.5), can induce acid-induced conformational changes in IAV, leading to viral inactivation before receptor binding [45]. However, IAVs adapted to humans, such as the pdm09 H1N1 virus, have HA proteins with lower activation pH, enhancing their stability and transmissibility in the human respiratory tract [46]. The pdm09 H1N1 virus adapted to human hosts from pigs, and it was suggested that an HA activation pH of 5.5 or less is one of the requirements of pandemic IAVs [38]. Indeed, the HA activation pH of human-adapted IAVs tends to range from pH 5.0 to 5.5, in contrast to animal-origin IAVs retaining a broader range of HA activation pH [39,47]. Therefore, the host-specific optimality in acid stability of IAV seems to be deeply associated with viral transmission and replication in the host [39].

### 2.4. Permissiveness to a Novel Host

Mutations in both the HA and NA proteins, crucial for virus entry and budding, as well as mutations in the viral RNA polymerase, are essential for the adaptation of avian IAVs to human hosts [48]. Among these adaptations, the glutamate-to-lysine substitution at position 627 (E627K) or the aspartate-to-asparagine substitution (D701N) in the PB2 protein have been well-documented as viral adaptive mutations for mammalian hosts [48,49,50,51,52]. The 2009 H1N1 pandemic IAVs, resulting from multiple reassortment events between avian, swine, and human IAVs, retained 627E and 701D residues but acquired alternative mutations in the PB2 protein (G590S and Q591R) in the vicinity of residue 627 [53]. These mutations enabled efficient replication of pdm09 in both humans and pigs, playing a pivotal role in viral adaptation. Additionally, several other mutations in the polymerase genes contribute to the fine-tuning of viral host specificity [54,55,56,57,58]. The NS1 protein, an antagonist of the host’s innate immune response, is also known to be one of the determinants affecting viral replication, adaptation, and transmission in a novel host [59,60,61,62]. Specifically, the C-terminal region of the NS1 protein contains the PDZ-binding motif. Sequence differences in the PDZ-binding motif, dependent on the host of origin, are recognized as significant virulence determinants of IAV [61,62].

These host-determining mutations can interfere with the interaction between viral proteins and host factors differing by species. The acidic nuclear phosphoprotein-2 (ANP32) family of proteins is regarded as key species-specific host factors. ANP32 proteins form a complex with the viral polymerase complex and enable efficient cRNA-to-vRNA replication in a species-specific manner [48,63,64,65]. RNA polymerases from avian IAVs replicate their genome inefficiently due to species-specific differences in ANP32. However, host-adaptive mutations, such as E627K in the PB2 protein, allow avian IAV to replicate efficiently in the presence of human ANP32A [63,64]. In addition, several adaptive mutations in vRNA modulate the sensitivity of IAVs to certain human interferon-stimulated genes (ISGs), including RIG-I and MxA, overcoming the human innate immune response for efficient viral replication [66,67,68].

## 3. Animal Hosts of IAV Raising a Zoonotic Potential

### 3.1. Pigs

Pigs play a crucial role as hosts for influenza A viruses, acting as mixing vessels [69]. However, solid evidence supporting the mixing vessels model is somewhat lacking. Generally, sialic acid receptor distribution in the swine respiratory tract is similar to that of humans. The distribution of α2,6-SA glycan is higher than that of α2,3-SA glycan in the upper respiratory tract of pigs, and both glycans are comparably distributed in the lower respiratory tracts such as bronchiole and alveolar regions [70,71].

Throughout history, co-infections of human, swine, and avian IAVs in pigs have resulted in the emergence of multiple reassortment viruses. Notably, the 2009 H1N1 pandemic virus (pdm09) originated from such reassortments [72,73]. The H1N1/pdm virus has successfully adapted to humans and now causes seasonal epidemics, underscoring the pivotal role of pigs as intermediate hosts in IAV pandemics [72,74,75].

The optimum HA activation pH of swine IAVs is known to retain a broader range, ranging from pH 5.0 to 6.3 [47,76]. The early H1N1/pdm virus, A/California/04/2009, showed a fusion pH of 5.5, while a more recent H1N1 virus mediated membrane fusion at pH 5.3 [38]. This suggests that a reduction in fusion pH may be necessary for further adaptation of swine IAVs to actively spread among human populations. In the context of mutations for replication in mammalian hosts, swine IAVs seem to utilize a similar but distinctive adaptation pathway compared to human IAVs. The majority of swine IAVs do not contain the E627K mutation in the PB2 protein [54]. Instead, swine IAVs possess distinctive adaptive mutations in PB2 (G590S/Q591R), which compensate for the lack of the E627K mutation [53,54,77]. Notably, these swine-origin IAVs containing the alternative mutations caused the 2009 H1N1 pandemic in humans, and the viruses established themselves in humans, becoming one of the seasonal flu strains [58,74,77,78]. Therefore, the adaptive mutations observed in pigs might have interchangeably contributed to viral adaptation in humans.

Zoonotic and reverse-zoonotic events occasionally occur between humans and pigs [79,80,81,82]. Currently, H1N1, H1N2, and H3N2 subtypes of influenza A viruses (IAVs) are globally endemic in pigs [83]. These viruses exhibit high genetic diversity, resulting from diverse hemagglutinin (HA) lineages, neuraminidase (NA), and other gene segments. It is predicted that the bi-directional transmission of IAVs between humans and pigs has contributed to this variability [80,81,82,84]. The 2009 H1N1 pandemic lineage (H1N1/pdm) initially spilled over to pigs, where it underwent genetic and antigenic diversification before reintroducing to humans. Importantly, swine IAVs have evolved distinctly from the human circulating virus, posing challenges to pandemic preparedness [82]. Beyond human-to-pig interfaces, instances of bird-to-pig transmission of IAVs have also been observed sporadically. Various subtypes such as H4N6, H3N2, H9N2, H5N1, H7N9, H10N8, H4N1, H6N6, H4N8, H5N2, and H7N2 have been reported to transmit to pigs [83,85]. Among these, H5N1 and H9N2 avian IAVs have exhibited multiple spillover events from avian species, displaying molecular markers indicative of mammalian adaptation. Therefore, active surveillance of novel IAV introductions in swine populations should be conducted to monitor potential cross-species transmission to humans through pigs.

### 3.2. Galliformes

Genus Galliformes comprises heavy-bodied, ground-feeding birds often reared by humans for their meat and eggs. They have been closely associated with humans for thousands of years. Due to high-density rearing systems, they are immunologically prone to infectious diseases, increasing the risk of transmission of more virulent pathogens [86]. These pathogens can occasionally cross host-specific barriers, infecting people who are closely associated with them, including veterinarians and farm or slaughterhouse workers [87].

The Galliformes and the Anseriformes (waterfowl) belong to the superorder Galloanserae and have a close genetic relationship. In the context of sialic acid receptors, however, they show a distinctive distribution. While both ducks and chickens express α2,3-SA and α2,6-SA glycans in multiple organs, α2,6-SA glycans are dominant in chicken tracheas, whereas α2,3-SA glycans are most dominant in duck trachea [19,21,88]. Similarly, other Galliformes, such as pheasants, turkeys, quails, and guinea fowls, show both α2,3-SA and α2,6-SA in the respiratory tract [20]. Therefore, they may play a crucial role as “mixing vessels,” facilitating the cross-species transmission of IAVs to humans. Diverse subtypes of avian IAVs have been transmitted from wild waterfowl to gallinaceous birds, and the viruses have adapted to the novel hosts [89,90].

Some H9N2 viruses found in poultry displayed a consistent preference for binding to human-like receptor analogs, in contrast to H9N2 viruses isolated in wild birds [91,92]. Moreover, some H9N2 avian IAVs showed relatively low fusion pH ranging between pH 5.4 and 5.85 [93]. Similarly, Zhong et al. suggest that an essential residue (363K) in HA for aerosol transmission of H9N2 viruses in chickens is responsible for high acid stability [94]. These findings support the potential role of gallinaceous birds as intermediate hosts for the spread of avian IAVs to humans. In contrast, the frequent generation of HA glycosylation during viral adaptation to gallinaceous birds can concomitantly induce stalk truncation of NA to balance HA and NA activity, which may be deleterious for viral transmission in humans [95]. Since there are some conflicting data, in-depth studies are necessary to evaluate the role of gallinaceous birds in the cross-species transmission of IAVs to humans.

IAV transmissions from waterfowl to gallinaceous birds are frequent. H1 to H11 subtypes of avian IAVs (except H13N2 IAV isolated in Turkey in 1990 and 1991) have been reported to circulate in gallinaceous birds, some of which crossed host-specific barriers and were transmitted to humans [87]. Among them, H5 or H7 highly pathogenic avian IAV (HPAI) transmission events to humans are the most noticeable due to their high morbidity and mortality in gallinaceous birds [96,97]. Since the first human infection case in 1997 by H5N1 HPAI, more than 890 cases of HPAI H5N1 human infection have been reported with approximately 50% case fatality proportion [87,96,98,99]. Other HPAIs, including H5N6, H7N7, and H7N9, have also been reported to cause human infection [100,101,102]. According to epidemiological studies, direct exposure to infected gallinaceous birds is thought to be the primary risk factor for these sporadic human infections with HPAI [103]. Low pathogenic avian IAVs, including H3N8, H6N1, H9N2, and H10N8, have also caused spillovers into humans from gallinaceous birds [104,105,106,107]. These spillovers are mostly limited to dead-end infections, but continuous monitoring of poultry-to-human infection of avian IAVs and efforts to break the transmission chain would be important to prevent the emergence of the next pandemic by IAVs.

### 3.3. Companion Animals

Companion animals are often considered family members and have close physical contact with humans [108]. Although the mutually beneficial relationship can positively impact the mental, physical, and social health of both, the close relationship can increase the risk of cross-species transmission of zoonotic pathogens [109]. Dogs, the first domesticated animals, have shared a profound bond with humans for over 30,000 years. Within the respiratory tract of dogs, both α2,3-SA and α2,6-SA coexist despite the prevalence of α2,3-SA [110]. Dogs exhibit susceptibility to IAV infection, including avian (H3N2, H5N1, H5N2, H6N1, H7N9, H9N2, and H10N8), human (H1N1/pdm and H3N2), and equine (H3N8) origin [6,111,112,113,114,115]. Two distinct canine IAVs (H3N8 and H3N2) have been established in this species, believed to have originated from equine and avian IAVs, respectively [111,116]. While these viral lineages have maintained stability in canine populations without reported zoonotic events, the H3N2 canine IAV has undergone evolutionary adaptations, resulting in enhanced mammalian adaptation, posing an increased risk as a potential zoonotic IAV [109,115,117,118,119,120,121,122]. Canine influenza viruses have acquired mutations that are related to human-like receptor preference, acid stability, and viral polymerase activity, although no well-known adaptive mutations such as E627K or D701N have been observed so far [117]. Moreover, reassortment with H1N1/pdm detected in Korea implies that co-infection of canine IAV with human viruses may lead to a novel virus with pandemic potential [112]. Therefore, cross-species transmission of animal-origin IAVs to dogs might enable the mammalian adaptation of these viruses. This underscores the potential role of dogs in the zoonotic transmission of IAVs, given their intimate association with humans [115].

Cats exhibit the presence of both α2,3-SA and α2,6-SA in their respiratory tract, rendering them susceptible to IAV infection. Various strains of avian (H5N1, H5N6, H7N2, and H9N2), canine (H3N2), and human (H1N1/pdm and H3N2) IAVs have been transmitted to cats [113,123,124,125,126]. While cases are mostly subclinical, some have proven to be lethal. Recently, the clade 2.3.4.4b HPAI has disseminated globally, and it has recurrently spilled over to cats in several countries, resulting in significant disease fatality [127,128,129]. In most cases, the viruses isolated from infected cats acquired mammalian adaptive mutations, such as E627K or D701N in the PB2, without altering their receptor specificity from α2,3-SA to α2,6-SA [125,126,127,129]. However, the potential role of cats as an intermediate host of IAVs warrants thorough investigation.

### 3.4. Minks

Minks and ferrets belong to the family Mustelidae, and they are genetically close [130]. Similar to ferrets, which have been widely used as animal models for influenza virus studies for several decades, minks are susceptible to several human respiratory viruses, including influenza A virus and SARS-CoV-2 [131]. α2,6-SA is predominantly present in the mink respiratory tract, but α2,3-SA is also found in the mink bronchioles, suggesting potential as an intermediate host for IAV [5,132]. Minks are raised in large groups for their fur; therefore, the consequences of cross-species transmission might be worse than those associated with other wild animals. Genetic analysis of IAVs isolated in minks found that several mammalian adaptive mutations were acquired during adaptation in minks [5]. Notably, according to the analysis using two public influenza databases, minks showed higher α-diversity than other mammalian species, meaning they were infected by more subtypes of IAVs [133]. A serological survey of farmed minks found that they are commonly infected with human H3N2, H1N1/pdm, and avian H7N9, H5N6, and H9N2 IAVs [5,134,135]. Animal experiments showed that minks are susceptible and permissive to human and avian IAVs, and aerosol transmission was stably observed in human IAVs [135]. Since 1984, several subtypes of IAVs have been detected in minks, including H1N1, H1N2, H3N2, H5N1, H5N6, H9N2, and H10N4 [5,132,136,137,138,139]. Of note, clade 2.3.4.4b H5N1, which has spread globally and caused a panzootic, has been found in farmed minks [140]. This virus showed evidence of mink-to-mink transmission and exhibited direct contact and airborne transmission in a ferret study [141]. Collectively, these findings highlight the significant role minks may play in the transmission and evolution of IAV, underscoring the need for stringent surveillance and biosecurity measures in mink farming to prevent potential zoonotic outbreaks.

### 3.5. Marine Mammals

Different subtypes of IAVs (H1, H3, H4, H5, H7, H10, and H13) have been detected in marine mammals for decades [142,143,144,145,146,147]. The seal pulmonary distribution of α-2,3-SA and α-2,6-SA revealed that α-2,6-SA receptors are widely expressed and α-2,3-SA receptors are also observed, but less frequently [142]. Mammalian adaptive mutations such as D701N in PB2 were found in the H3N8 seal-adapted IAVs in 2011, but similar mutations were not found in other seal-adapted IAVs [142,145]. In ferret studies, the H10N7 virus isolated in seals preferred α-2,6-SA and was aerosol transmissible [146]. Substitutions found in the seal-adapted HA lowered the fusion pH and changed the receptor binding pattern, suggesting that seal infection with avian-origin IAV could lead to zoonotic transmission [146]. Of note, the 2009 pandemic H1N1 viruses were detected twice in northern elephant seals in 2010 and 2019 in California [147,148]. The phylogenetic analysis revealed that these isolated viruses are genetically close to contemporary pdm09 H1N1 viruses circulating in the United States. These events indicate that reverse zoonosis from humans could occur in marine mammals, similar to those in pigs. More recently, several species of marine mammals were found to be infected with clade 2.3.4.4b HPAI in South America, along with the HPAI outbreaks on the continent during 2022–2023 [149]. Sequencing analysis of the viruses isolated from the infected animals found that the virus had already acquired several mammalian adaptive mutations (Q591K and D701N mutations in the PB2 gene). Collectively, these findings indicate that marine mammals are susceptible to both avian and human IAVs, emphasizing the potential role of marine mammals as intermediate hosts in IAV cross-species transmission.

### 3.6. Other Mammalian Species

Other mammalian species are susceptible to avian IAVs. There have been several spillovers of avian IAVs into zoo animals, including tigers, leopards, and lions [6]. The clade 2.3.4.4b H5N1 viruses have spread along migratory flyways, resulting in genetic reassortment with local low pathogenic avian influenza viruses (LPAIs), leading to an unusually high propensity to infect mammals [150,151]. Notably, independent spillovers of the clade 2.3.4.4b H5N1 virus into red foxes occurred in 2021 and 2022 [152]. The virus isolated from the infected red foxes acquired the E627K mutation in PB2, but no further transmission between foxes was observed.

In March 2024, milk and tissue samples from dairy cattle tested positive for IAV, which was further characterized as the clade 2.3.4.4b H5N1 virus [128]. The virus infection caused mastitis in the infected cows, and evidence of cow-to-cow transmission was found [128]. In addition, domestic cats that consumed raw colostrum and milk from sick cows exhibited a fatal systemic influenza infection [128]. The viruses isolated from the infected cat were phylogenetically related to those isolated from the cows. More importantly, cow-to-human transmissions were confirmed by monitoring people exposed to the infected cows or contaminated materials including cow’s milk [153,154,155]. Phylogenetic analysis revealed that this spillover was the result of a single interspecies transmission event, preceded by genetic reassortment between the clade 2.3.4.4b H5N1 and other LPAIs in wild bird species [150]. This robust genetic reassortment within highly permissive hosts increases genetic diversity and potentially contributes to interspecies transmission [156]. Some mammalian adaptive mutations, including E627K or D701N in PB2, were detected at low frequencies in the bovine H5N1 IAVs [150]. Both α-2,6-SA and α-2,3-SA glycans are expressed in the mammary gland, respiratory tract, and cerebrum of dairy cows, which might explain their susceptibility to IAVs [157]. This suggests a novel route of cross-species transmission of IAVs through the novel animal hosts, dairy cows.

## 4. Conclusions

Cross-species transmission of viruses from wild or domesticated animals is one of the main sources of emerging infectious agents in humans. Frequent outcomes of cross-species transmission are asymptomatic infection or dead-end events where the virus fails to establish in a new host species. However, rare but critical viral evolution in a new host can cause significant adaptations, leading to widespread outbreaks. From their natural reservoirs in wild waterfowl, IAVs have been transmitted to terrestrial birds and mammals, including humans (Figure 1). Viral adaptation in new hosts or genetic reassortment between human- and animal-origin IAVs in “mixing vessels” can lead to unpredictable changes, increasing zoonotic potential.

Since the 2020s, cross-species transmission of avian-origin IAVs to mammals, including humans, has been increasingly reported. Improved IAV detection and sequencing technologies contribute to this trend, but the risk of stable human-to-human transmission via intermediate hosts remains high. Active surveillance and assessment of zoonotic risks in various animal hosts are crucial for monitoring and preventing the next influenza pandemic.

## Figures and Tables

**Figure 1 viruses-16-01129-f001:**
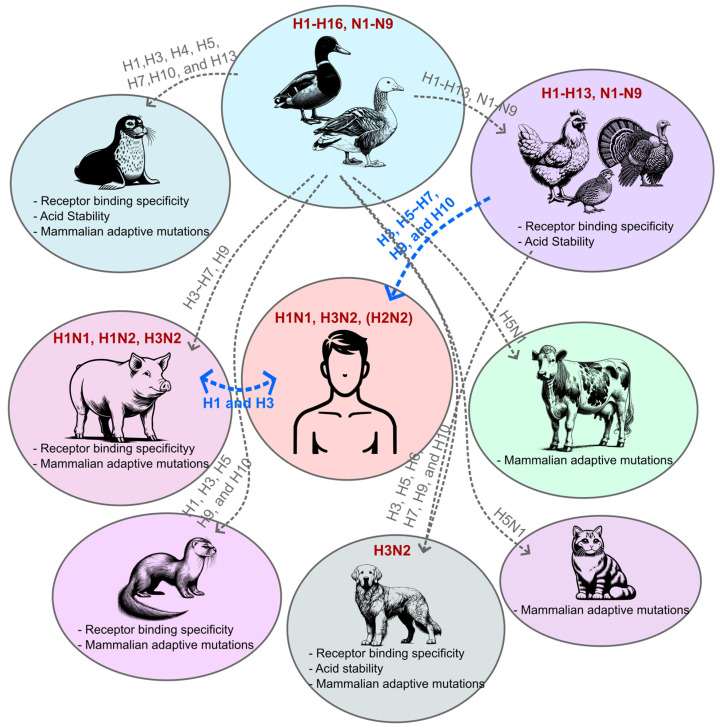
Cross-species transmission of influenza A virus. From their natural reservoirs in wild waterfowl, influenza A viruses (IAVs) have been transmitted to various animal species. The majority of these cross-species transmission events result in dead-end interactions between the novel IAVs and the animals (grey dashed lines, H5N1 in dairy cattle requires further studies), but some subtypes successfully circulate within that animal population (bolded red). These cross-species transmissions can also occur between animals and humans, raising public health risks (blue dashed lines). The host-adapted changes in animal-isolated IAVs are summarized below the host species.

## Data Availability

Not applicable.

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
