# Peer review of "Exploring Potential Intermediates in the Cross-Species Transmission of Influenza A Virus to Humans"

_viruses, 2024, doi:10.3390/v16071129_

Round 1

Reviewer 1 Report

Comments and Suggestions for Authors

Major comments:

The paper is missing a Methods section, which is essential for any systematic review. Please visit the Cochrane or PRISMA guidelines and draft a methods section to accompany your excellent review. What were your inclusion/exclusion criteria, how many articles were screened, how many were excluded, what was the article review protocol, how many reviewers reviewed articles, and how were disagreements resolved between reviewers? These are just a few of the statements that will be needed before this paper can be considered for publication. Please see other systematic reviews published in Viruses to identify the steps needed to qualify for a systematic review.

Minor comments:

Other than the above, this is one of the most well written papers I have reviewed in a long time. The author clearly is a great writer and understands the intricacies of inter-species transmission of IAVs, particularly those related to the sialic acid receptors for virus binding across species. I am not sure how novel these findings are, but I do appreciate that the author took the time to sift through the literature and write a well-composed summary of the findings.

Line 17-18, 95-96, 117-119, 229-230, 262, : references?

Author Response

Comment 1: 

The paper is missing a Methods section, which is essential for any systematic review. Please visit the Cochrane or PRISMA guidelines and draft a methods section to accompany your excellent review. What were your inclusion/exclusion criteria, how many articles were screened, how many were excluded, what was the article review protocol, how many reviewers reviewed articles, and how were disagreements resolved between reviewers? These are just a few of the statements that will be needed before this paper can be considered for publication. Please see other systematic reviews published in Viruses to identify the steps needed to qualify for a systematic review.

Response: 

We greatly appreciate the reviewer’s suggestion. However, we would like to clarify that this manuscript is a review article, not a systematic review. To the best of our knowledge, review articles are more flexible and less structured compared to systematic reviews and do not require a predefined protocol.

Minor comments:

Other than the above, this is one of the most well written papers I have reviewed in a long time. The author clearly is a great writer and understands the intricacies of inter-species transmission of IAVs, particularly those related to the sialic acid receptors for virus binding across species. I am not sure how novel these findings are, but I do appreciate that the author took the time to sift through the literature and write a well-composed summary of the findings.

Line 17-18, 95-96, 117-119, 229-230, 262, : references?

We really appreciate for the reviewer’s valuable comments. We have added the references as the reviewer suggested.

Reviewer 2 Report

Comments and Suggestions for Authors

The author summarized the mechanisms of cross-species transmission of influenza A virus to humans and a zoonotic potential of animal hosts of IAV. Now that the infection of H5N1 IAVs to cattle has been confirmed, this review will be published at good timing because it summarizes the transmission of the virus from wild birds to other hosts. My comments on the improvements are listed below.

lines 29-33

This paragraph is not fully explained. It is unclear whether this paragraph is talking about adaptation and evolution in a new host or the emergence of a pandemic virus due to reassortants in intermediate hosts.

line 116

Although this section summarizes permissiveness to a new host by adaptive mutations in PB2, NS1 has long been thought to be involved in host adaptation by inhibition of the innate immune system. Although there may be no clear evidence, it would be better to summarize and describe host selection for NS1 as well.

line 307

It should be mentioned that reassortants within IAV of wild birds occurred prior to infection of cattle, and that reassortants not only between IAVs of different hosts but also IAVs within wild birds may be a factor in transmission of different host.

line 697-701

Quotes seem to be from biorxiv, but the name of the journal and other information are missing.

Author Response

The author summarized the mechanisms of cross-species transmission of influenza A virus to humans and a zoonotic potential of animal hosts of IAV. Now that the infection of H5N1 IAVs to cattle has been confirmed, this review will be published at good timing because it summarizes the transmission of the virus from wild birds to other hosts. My comments on the improvements are listed below.

Comments 1: 

lines 29-33

This paragraph is not fully explained. It is unclear whether this paragraph is talking about adaptation and evolution in a new host or the emergence of a pandemic virus due to reassortants in intermediate hosts.

Response: 

  • We appreciate for the reviewer’s comment. We have combined the paragraph to the previous paragraph and have revised it to focus more on reassortment events in intermediate hosts (Lines 28-31).
  • “For example, the 2009 H1N1 pandemic virus (pdm09) emerged through reassortment events between human, swine, and avian IAVs [2, 4]. The pigs played a crucial role in the genesis of pdm09 and its transmission to humans, underscoring the significant role of intermediate hosts in the emergence of new pandemics.”

Comments 2: 

line 116

Although this section summarizes permissiveness to a new host by adaptive mutations in PB2, NS1 has long been thought to be involved in host adaptation by inhibition of the innate immune system. Although there may be no clear evidence, it would be better to summarize and describe host selection for NS1 as well.

Response:

  • We greatly appreciate the reviewer’s suggestion. We have described the role of NS1 protein in host selection (Lines 124-129)
  • “The NS1 protein, an antagonist of the host's innate immune response, is also known to be one of the determinants affecting viral replication, adaptation, and transmission in a novel host [59-62]. Specifically, the C-terminal region of NS1 protein contains the PDZ-binding motif. Sequence differences in the PDZ-binding motif, dependent on the host of origin, are recognized as significant virulence determinants of IAV [61, 62].”

Comment 3:

line 307

It should be mentioned that reassortants within IAV of wild birds occurred prior to infection of cattle, and that reassortants not only between IAVs of different hosts but also IAVs within wild birds may be a factor in transmission of different host.

Response:

  • We appreciate for the reviewer’s suggestion. We have revised the paragraph to contain genetic reassortments within wild bird species occurred prior to spillover into cattle, and the reassortment within highly permissive hosts may contributes to interspecies transmission of IAVs (Lines 313-333).
  • “Other mammalian species are susceptible to avian IAVs. There have been several spillovers of avian IAVs into zoo animals, including tigers, leopards, and lions [6]. The clade 2.3.4.4b H5N1 viruses have spread along migratory flyways, resulting in genetic reassortment with local low pathogenic avian influenza viruses (LPAIs), leading to an unusually high propensity to infect mammals [150, 151]. Notably, independent spillovers of the clade 2.3.4.4b H5N1 virus into red foxes occurred in 2021 and 2022 [152]. The virus isolated from the infected red foxes acquired the E627K mutation in PB2, but no further transmission between foxes was observed.
  • In March 2024, milk and tissue samples from dairy cattle tested positive for IAV, which was further characterized as the clade 2.3.4.4b H5N1 virus [128]. The virus infection caused mastitis in the infected cows, and evidence of cow-to-cow transmission was found [128]. In addition, domestic cats that consumed raw colostrum and milk from the sick cows exhibited a fatal systemic influenza infection [128]. The viruses isolated from the infected cat were phylogenetically related to those isolated from the cows. More importantly, cow-to-human transmissions were confirmed by monitoring people exposed to the infected cows or contaminated materials including cow’s milk [153-155]. Phylogenetic analysis revealed that this spillover was the result of a single interspecies transmission event, preceded by genetic reassortment between the clade 2.3.4.4b H5N1 and other LPAIs in wild bird species [150]. This robust genetic reassortment within highly permissive hosts increases genetic diversity and potentially contributes to interspecies transmission [156].”

Comment 4:

Quotes seem to be from biorxiv, but the name of the journal and other information are missing.

Response:

  • Are you referring to line 727, reference 148 Kristensen, C.; Jensen, H. E.; Trebbien, R.; Webby, R. J.; Larsen, L. E., 2024)? We have added more information to the reference list. (Lines 763-764)